# Can increased dietary fibre level and a single enrichment device reduce the risk of tail biting in undocked growing-finishing pigs in fully slatted systems?

**Jen-Yun Chou**[1,2,3][◉]\*, **Keelin O'Driscoll**[1][◉], **Dale A. Sandercock**[2][◉], **Rick B. D'Eath**[2][◉]

**1** Pig Development Department, Teagasc, Fermoy, Co. Cork, Ireland, **2** Animal & Veterinary Sciences Research Group, Scotland's Rural College, Easter Bush, Midlothian, United Kingdom, **3** Royal (Dick) School of Veterinary Studies, University of Edinburgh, Easter Bush, Midlothian, United Kingdom

◉ These authors contributed equally to this work.
\* jenyun.chou@gmail.com

## Abstract

This study evaluated the effectiveness of combined dietary and enrichment strategies to manage tail biting in pigs with intact tails in a conventional fully-slatted floor housing system. A 2 × 2 × 2 factorial design was used. Pigs had either a high fibre (weaner 5.3% and finisher 11.6% of crude fibre) or standard fibre diet (weaner 3.7% and finisher 5.9% of crude fibre). In the weaner stage, pigs had either a spruce wooden post (supplied in a wall-mounted dispenser) or a rubber floor toy as a enrichment device, and in the finisher stage, they had either the same or alternate enrichment item. Six hundred and seventy-two pigs were assigned to 48 pens of 14 pigs and followed from weaning until slaughter. Individual tail lesion scores and pen level behaviours were directly recorded every 2 weeks. Twenty-six pens had tail biting outbreaks and 161 injured pigs needed removal for treatment. Pigs fed with the high fibre diet performed more tail biting ($p < 0.05$) and tended to have a worse tail damage scores than those fed the standard fibre diet ($p = 0.08$). Pigs which had the floor toy as weaners and wood as finishers tended to have fewer tail lesions in the finisher stage than their counterparts ($p = 0.06$). Pigs receiving the floor toy as enrichment interacted with the enrichment more frequently overall ($p < 0.001$) and performed fewer harmful behaviours in the weaner stage ($p < 0.05$). Overall, higher fibre in the diet in a relatively barren environment did not help reduce tail biting or tail lesions. Altering the fibre level in the pigs' diet and providing a single enrichment device to undocked pigs on fully slatted floors resulted in a high level of tail biting and a large proportion of pigs with partial tail amputation.

## Introduction

EU Commission Directive 2001/93/EC [1] lays down minimum standards for pig welfare, stating that tail docking is banned as a routine practice to control tail biting. However, according to a recent survey, among all EU countries approximately 77% of pigs are still tail docked [2].

**Data Availability Statement:** All relevant data are within the manuscript and its Supporting information files.

**Funding:** This project was co-funded by the Teagasc Walsh Fellowship, the Department of Agriculture, Food and the Marine in Ireland, and Scotland's Rural College (SRUC). SRUC also receives funding from the Rural & Environmental Science & Analytical Services Division of the Scottish Government.

**Competing interests:** The authors have declared that no competing interests exist.

One of the main challenges identified was pig producers' concern about the consequences of tail biting if docking is not performed. Indeed Lahrmann et al [3] recorded a higher prevalence of tail lesions among undocked than docked pigs when housed on conventional partly-slatted floor systems. However, using tail docking to manage tail biting masks the underlying issues caused by insufficiencies in the production system [4,5].

Studies have suggested that the floor type (solid or slatted) is an important risk factor for tail biting; slatted floors are associated with a higher prevalence of tail biting [6,7]. However, floor type is often confounded with other factors such as enrichment provision. Although the absence of straw and other bedding material has been identified as the most vital factor influencing tail biting prevalence [6,8,9], on fully-slatted floors loose material hinders slurry management [10], so cannot be used without restraint. Therefore, tail biting is especially difficult to manage in fully-slatted systems, which are commonly utilised in EU pig production systems [11], without major changes that could have negative economic implications for the producers [5]. This is also the main reason why there is still a high prevalence of tail docking, since other economically feasible solutions to practically reduce the risk of tail biting are lacking.

Wood was listed as a suboptimal organic environmental enrichment material in a European Commission recommendation on how to manage pigs without docking their tails [12]. The species of wood can affect the pigs' rate of interaction with it; studies have shown that spruce (*Picea sitchensis*) as a softer species can attract more interactions from the pigs compared to other species that are harder [13,14]. However, although wood is an organic material, it does not always generate more interactions from pigs than inorganic enrichment materials [15], and the way an enrichment material is presented can also significantly influence its attractiveness to pigs [16]. Nevertheless, Telkänranta et al [17] showed that provision of branches of fresh cut wood reduced tail lesions in undocked pigs, compared to polyethene pipes and chains. A recent survey reveals that wood is the most favoured organic material by Irish pig farmers to be used as an enrichment [18]. Other forms of inorganic point-source enrichment materials have also been tried on undocked pigs but were less effective in preventing tail biting [19,20]. In a previous study, we have shown that when either a spruce post or a rubber floor toy was provided as a single enrichment item in a pen of seven tail docked finishing pigs, they attracted a similar level of interactions from pigs [14]. Therefore, these two items were selected as the enrichment materials to be further tested in the current study, using undocked pigs, and through the entire production cycle.

Studies have shown that dietary modification can also have an effect on tail biting behaviour in pigs particularly in terms of protein level and mineral content [9]. The effect of dietary fibre on pig behaviour has mainly been investigated in relation to satiety among restricted-fed sows [21]. In these studies, the authors suggested that provision of a fibre source with high bulkiness could reduce oral manipulation behaviours immediately post-feeding, and highly fermentable fibre could further reduce activity levels by prolonging satiety. In growing pigs, Kallabis and Kaufmann [22] found that fattening pigs fed higher quantities of dietary fibre tended to have fewer meals per day and a lower daily feed intake but spent more time feeding and ate more slowly. Although these authors recognised the effect of high dietary fibre on improving satiety and inferred a likely reduction in oral manipulative behaviours, evidence for this is still lacking in the scientific literature. A recent study explored the effect of high dietary fibre in weaner pigs' diet from 4 to 7 weeks of age on tail biting, using a commercially formulated high fibre diet [23]. The pigs also received a small amount of straw as environmental enrichment. They did not find any effect of high fibre diet on the weaners' tail biting behaviour; however, this could be due to a small difference in the fibre level between diets, and the short period of time during which the high fibre diet was provided.

The present study combined the use of a point-source enrichment material that is compatible with slatted-floor systems, and a dietary modification of increased fibre content from 4 weeks post-weaning until slaughter to evaluate their effectiveness at controlling tail biting in undocked pigs. For the enrichment treatment, an inorganic rubber enrichment device was compared with a spruce post. As rooting is an important behaviour for pigs [24], both types of enrichment device were presented on the floor in ways that prevented them from being soiled easily. In terms of dietary fibre, soybean hulls with moderate fermentability, water-holding capacity and bulkiness [21] were used as the main ingredient to increase fibre level. It was hypothesised that groups with the higher fibre diet and the wooden post as enrichment would have a lower occurrence of tail biting behaviour and lesions.

## Material and methods

### Ethical considerations

All procedures in the experiment were approved by the Teagasc Animal Ethics Committee (TAEC124/2016).

Due to the high risk of tail biting outbreaks occurring, we took several steps to minimise the negative impacts on pig welfare. This included protocols for tail injury inspection and treatment, and for outbreak intervention.

All pens were routinely checked for tail biting incidents at least three times per day by the experimenter and twice daily by the farm staff (with a reduced inspection frequency of around three times on Sunday). Antibiotics (cutaneous spray Alamycin® Aerosol, Norbrook and subcutaneous injection of Norocillin®, Norbrook) and analgesics (subcutaneous injection of Loxicom®, Norbrook) were administered when pigs' tails were swollen with signs of infection. Pigs were removed temporarily to a hospital pen for tail treatment whenever necessary to aid recovery. This was the case when a bitten tail did not heal within 3 days, or if the tail became infected or inflamed (identified by redness and swelling. Pigs under medical treatment stayed in the hospital pens until healed, but no more than 14 days to facilitate reintroduction to the home pen [25]. If an animal had sustained severe injuries which required prolonged recovery time, it was removed from the study and humanely euthanised when necessary as evaluated by the experimenter. Any movement of animals between home and hospital pens was monitored and recorded in detail using pigs' individual ear tag identifications.

The outbreak definition and intervention protocol used was described in detail in a previously published study [25]. In short, an outbreak was defined when (a) three or more pigs per pen with fresh blood on their tails that can be identified from outside the pen, (b) 1–2 pigs with bloody tails per pen for 72 h, or (c) three or more pigs with severe tail damage for 72 h without fresh blood present. The interventions involved either 1) removing victims, 2) removing biters or 3) adding additional enrichment (ropes). The selection of each intervention protocol to be used was carried out using a pre-defined randomised schedule. Chou et al [25] previously reported that there was no difference in the effectiveness of the three intervention strategies. In the current study, the intervention protocols were randomised across treatments, and there was no difference between experimental treatments in terms of the number of outbreaks. Therefore, we consider that the interventions applied did not impact on study outcomes.

### Animals and housing

A total of 672 piglets (Landrace × Large White) were farrowed from 58 litters in the experimental unit in Teagasc, Moorepark, Ireland, over two batches at an interval of 3 weeks. Piglets were teeth-clipped after farrowing, and their tails left undocked. No castration of male pigs

was carried out. A conventional farrowing pen system (2.4 m × 1.8 m) was used. Each pen contained a metal farrowing crate (2.2 m × 0.6 m) for the sow, and a floor heating plate (1.6 m × 0.4 m) for the piglets. The floor area was fully-slatted except for the heating plate, and the piglets had access to a nipple drinker and a rope was provided to the sow as enrichment. The room temperature during lactation was maintained at around 24˚C by the computerised control system. During lactation, sows were fed two to three times daily a standard pelleted diet (from 60 MJ of digestible energy at the start of lactation and gradually increased to 140 MJ at weaning). The staff checked the feed trough daily for sows' feed intake to adjust individual feed curves to ensure the intake was as close to *ad libitum* feeding as possible.

At weaning (4 weeks post-farrowing), all piglets were individually ear tagged for identification and weighed, and piglets lighter than 5 kg (17 in each batch) were not included further in the experiment. Remaining piglets were then randomly assigned into one of 6 blocks (three per farrowing batch), with each of the 8 treatments represented once in each block, forming 48 pens in total (14 pigs per pen). Pigs were balanced for weight between pens, and also for sex (half male/female) and litter mates within pens as far as practically possible. The average weaning weight was 7.60±1.26 kg. Pens within the same block for each treatment were located along the same corridor in the weaner and finisher housing. Throughout the production cycle, pigs were fed *ad libitum* by a single-spaced wet-dry feeder with dry pelleted feed per pen. There were two drinkers in the pen per 14 pigs, one nipple drinker inside the feeder and another nipple drinker separate to the feeder. Weaner pens were dimensioned 2.4 m × 2.6 m with a fully-slatted plastic floor. The ambient temperature was maintained by automatic heating and negative pressure mechanical ventilation at 28˚C in the weaner house immediately post-weaning. It was lowered by 2˚C every 2 weeks thereafter. Seven weeks post-weaning, pigs were transferred to the finisher facility (concrete slatted floors, dimensions of 4 m × 2.4 m), remaining in the same groups of 14. In the finisher house the temperature was maintained at 20˚C with the same ventilation system as in the weaner house, only without heating. All rooms were equipped with windows which enabled the pigs to be in contact with natural light. Artificial lighting (150 lux in weaner house and 130 lux in the finisher house) was provided for 10–12 hours/day to ensure sufficient lighting to retain a normal circadian rhythm.

## Experimental design

The study used a 2 × 2 × 2 factorial design. This was based on the type of enrichment in the weaner stage (wood or toy), finisher stage (wood or toy), and the level of fibre (standard or high) in the diet. Each treatment was replicated 6 times (n = 48; i.e. three times per batch).

**Enrichment treatments.** At weaning, half of the pens were each provided with a rubber floor toy (Easyfix Luna 117®, Easyfix, Ballinasloe, Ireland), while the other half were each given a dried spruce squared wooden post. The rubber floor toy was in a shape of a sphere in the centre with 12 arms stretching out at a length of 0.12 m (Fig 1A) and was provided loose on the floor. At the start of the experiment, the wooden posts were 1.100 ± 0.001 m in length, 1.342 ± 0.101 kg in weight and 0.231 ± 0.002 m in perimeter (a rectangular shape at around 0.08 ˣ 0.03 m; mean ± s.d.), and they were provided in a plastic dispenser (Funbar®, Jetwash Ltd., Carrigallen, Ireland; weaner 0.4 m and finisher 0.7 m in length, Fig 1B) fixed diagonally on the wall, angled at around 45˚. The lower end of the dispenser was at 0.3 m above the floor, and the upper and lower end of the wooden post was exposed for pigs' use. Upon transferring to the finisher house, half of the pens continued with the same type of enrichment and half of the pens alternated to the other type. No negative control (no enrichment) was used in this study as it is mandatory to provide environmental enrichment to pigs in the EU [1], and thus we considered that a meaningful experimental design in this context would be to compare

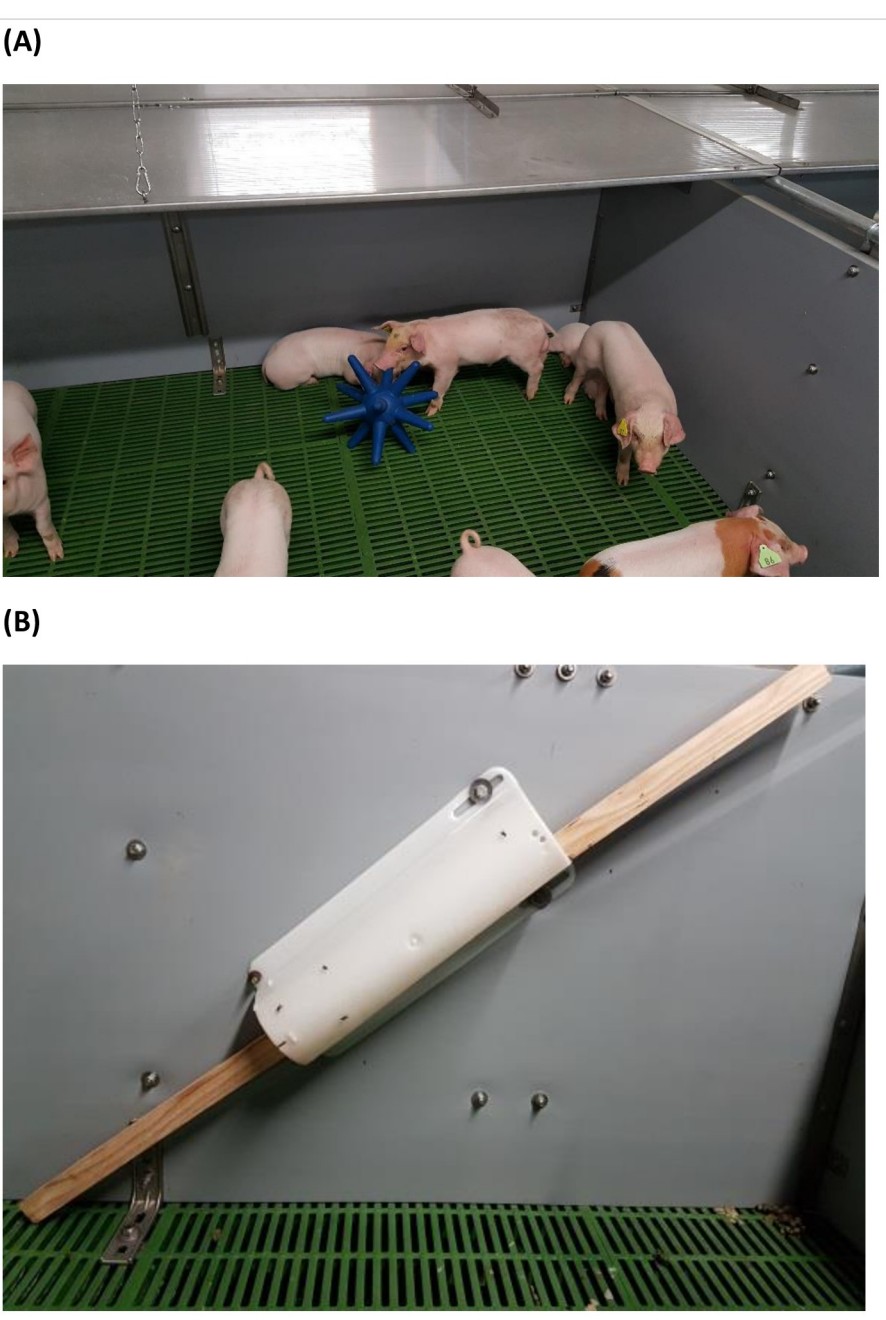

**Fig 1. Enrichment items.** (A) Rubber floor toy (Easyfix Luna 117®). (B) Spruce (*Picea sitchensis*) post provided in the Funbar® dispenser.

different minimal environmental enrichment provisions with a strict tail biting control protocol in place, which was done in the current study.

**Dietary treatments.** All weaners were given a commercial starter diet (Startrite 88, Provimi, Ireland) in the week immediately post-weaning, followed by a standard link diet (home-milled) for two weeks. The standard link diet was rich in lysine and amino acids, and this is a common practice on Irish farms to ensure the diets meet the nutritional requirements for pigs post-weaning. At the beginning of week 4 post-weaning, pigs in the standard fibre treatment

**Table 1. Formulation and chemical analysis of treatment diets in different stages.**

| | Weaner | | Finisher | |
|---|---|---|---|---|
| | Standard diet | High fibre diet | Standard diet | High fibre diet |
| **Ingredients (%)** | | | | |
| Wheat feed flour | 38.00 | 37.00 | 45.37 | 47.56 |
| Soybean meal (48% protein) | 25.00 | 25.00 | 13.50 | 13.50 |
| Maize | 15.36 | 10.01 | 5.89 | 5.80 |
| Barley | 18.00 | 9.70 | 24.00 | - |
| Soybean hulls | - | 10.45 | 8.50 | 26.30 |
| Wheat bran | - | 1.00 | - | - |
| Vegetable oil | 1.00 | 4.20 | 0.10 | 4.20 |
| Calcium carbonate | 1.25 | 1.25 | 1.25 | 1.25 |
| DL-Methionine | 0.08 | 0.08 | 0.08 | 0.08 |
| L-Lysine HCl | 0.44 | 0.44 | 0.44 | 0.44 |
| L-Threonine | 0.11 | 0.11 | 0.11 | 0.11 |
| Monocalcium phosphate | 0.37 | 0.37 | 0.37 | 0.37 |
| Sodium chloride | 0.30 | 0.30 | 0.30 | 0.30 |
| Vitamin and trace mineral mixture | 0.10 | 0.10 | 0.10 | 0.10 |
| **Chemical analysis (g/kg feedstuff)** | | | | |
| Dry matter | 885 | 888 | 883 | 888 |
| Crude ash | 45 | 47 | 40 | 45 |
| Crude protein | 196 | 193 | 142 | 154 |
| Crude fibre | 37 | 53 | 59 | 116 |
| Neutral detergent fibre | 106 | 142 | 146 | 240 |
| Acid detergent fibre | 45 | 74 | 78 | 155 |
| Acid detergent lignin | 8 | 9 | 10 | 14 |
| Digestible energy (kcal/kg dry matter)[a] | 3883.7 | 3796.8 | 3570.7 | 3275.8 |

[a] The digestible energy was calculated by the formula with the highest regression coefficient and lowest residual standard deviation of the prediction values described in Noblet and Perez [26].

continued to be fed with a diet containing a wheat, soybean and barley-based formula, typical of diets used in Irish systems (Table 1). In the high fibre diets, the addition of vegetable oil was used to match the energy density of the standard diets, whereas the carbohydrate ingredients such as maize and barley were mostly substituted with soybean hulls as the main fibre source (with some inclusion of wheat bran in the weaner diet). These steps were taken in order to keep the energy, protein, lysine and mineral level as closely similar as possible between the dietary treatments, while increasing the fibre level in the high fibre diet. The finishers' diets were formulated so that the high fibre diet was almost double the crude fibre of the standard diet, and around 1.5 times in the weaners' diets. All diets were milled on site, and pelleted. The feed samples were sent to an external laboratory (LKS, Niederwiesa, Germany) for a full chemical analysis (Table 1).

## Measurements

**Physical scores.** Tail and ear lesion scores were recorded by one experimenter for each individual pig in week 4 post-weaning and fortnightly thereafter until slaughter. Tail lesions were scored using a scoring system adapted from the FareWellDock project [25]. The severity of damage and the freshness of blood present were each scored on a 0 (optimal) to 3 (worst) scale. Based on findings of a positive correlation between the length of tail loss and risk of

**Table 2. Ethogram for direct behaviour observation.**

| Behaviours | Description of behaviour |
|---|---|
| Tail manipulation | Tail-in-mouth behaviour[a] on another pig which is not feeding |
| Tail manipulation at the feeder | Tail-in-mouth behaviour[a] on another pig which is feeding |
| Ear manipulation | Ear-in-mouth behaviour[a] on another pig |
| Manipulation of other body parts | Manipulation with mouth open of another pig in a body part other than tail and ear, e.g. face, snout, hock, or genital |
| Belly-nosing | Rubbing/manipulating a pen mate's belly/flank region with a rhythmic up-and-down movement |
| Mounting | Having two front legs on the back of a pen mate |
| Engaging in aggression behaviour | Pushing, aggressive biting, head-knocking, fighting with pen mates, excluding play-fighting, which does not involve a pig that is feeding or around enrichment. Play fighting is defined as when a pig shows signs of individual play when fighting with another pig. |
| Aggression behaviour at the feeder | Performing the above aggression behaviour during feeding or towards a pig that is feeding |
| Individual play | Scampering, pivoting, head tossing, flopping, and pawing [28,29], excluding social play |
| **Interactions with enrichment[b]** | |
| Biting device | Oral manipulation of the device with the mouth open |
| Rooting device | Nasal manipulation of the device by manoeuvring the device with the snout |
| Aggressive encounter | Biting, head-knocking or pushing other pigs over access to the device |
| Other | Physical contact with the device other than mouth or snout (limbs, body, etc.) |

[a] Tail-in-mouth or ear-in-mouth behaviour is defined as when a pig put the other pig's tail/ear in its mouth, with or without chewing or biting movement.

[b] If ropes were present in the pen due to outbreak control, they were recorded separately.

infections [27], if severely bitten tails were shortened, a tail amputation score was allocated based on a visually estimated level of amputation (0: no amputation, 1: remaining tail longer than around 50% of the original length, 2: remaining tail length between 25–50% of the original length, 3: remaining tail shorter than 25% of the original length). Ear lesions were scored on a 0–4 scale described in [13].

**Behaviour observations.** Behaviour observations were conducted by direct behaviour sampling continuously for 5 minutes at the pen level, based on a predefined ethogram (Table 2). The frequency of each behaviour was recorded by hand, and a behaviour bout that was longer than one minute was recorded as a new bout. Observations were carried out by one observer on two different days every other week from week 3 post-weaning. On each day there was one session in the morning between 10:00 h—13:00 h and another in the afternoon between 15:00 h—18:00 h (i.e. 4 recordings per pen per week). Before recording, the observer entered the house and walked down the passageway and back, announcing her presence with a vocal confirmation, so that all pigs could be aware of human presence and acclimatise. The observer was proficiently trained in conducting direct behaviour observation on pigs in previous studies, and practice sessions were also arranged before the actual data collection to ensure the quality of data collection.

**Production measures.** Individual pigs were weighed at weaning, when transferring to the finisher house and two days before slaughter. Feed intake was recorded daily by a computerised automatic recording system (BigFarmNet, Big Dutchman, Vechta, Germany) at the pen level. Cold carcass weights were obtained from the slaughterhouse report.

**Enrichment measurement.** To compare the rate of wear of enrichment items, the length and weight of the wooden posts were recorded before the start of the experiment, and subsequently every two weeks. The rubber floor toys were also weighed beforehand and every two weeks thereafter.

## Statistical analysis

Data were analysed using SAS Base 9.4 (SAS Institute Inc., Cary, NC, USA). Chi-squared tests were used to test if the likelihood of having a tail biting outbreak was greater in any treatment. A generalised linear mixed model with a Poisson distribution was used to analyse the number of pigs per pen that were removed and treated for tail injuries during the course of the experiment, including batch, enrichment and diet treatments, and all interactions as fixed effects, and block as random effect (n = 48).

Linear mixed models with a normal distribution were used to analyse continuous data including the proportion of each lesion score within a pen, behaviours, weight, feed intake and enrichment measurements. All residuals were checked for normality after model fitting. If non-normally distributed residuals were found, data were transformed using a square root transformation which generally improved normality of residuals. This was the case for all behavioural measures except enrichment interaction, all harmful behaviours combined and all behaviours combined. It was only for the proportion of tail damage score 2 & 3 in the finisher stage that transformation did not improve residual normality. Thus a generalised linear mixed model was used instead, with a Binomial distribution and a logit link function. Models for lesion scores and behaviours included fixed effects of batch, week, enrichment and diet treatments, and the interactions between treatments, with block as the random effect and week as the repeated effect. The data in the weaner and finisher stage were initially analysed separately, then also combined. Only the enrichment treatment in the weaner stage was included as a fixed effect for the weaner data, while for the finisher data, enrichment in both stages and their interactions were used. This was because treatment during the finisher stage could not account for the outcome in the weaner stage. When weaner and finisher data were combined, enrichment provided at the time of the recording was used. Week was always included as the repeated effect. The best-fit model was determined by step-wise removal of fixed effects comparing between Akaike's Information Criterion (AIC), Corrected Akaike's Information Criterion (AICC) and Sawa Bayesian Information Criterion (BIC, all criteria suggest a better model fit if smaller).

Lesion scores were analysed as the proportion of each score in a pen. Score 2 and 3 for tail and ear lesions were combined for analysis. Because removal of sick or injured pigs could reduce group sizes temporarily, behaviours were analysed as frequency per pig per minute. The data collected at different times of day and on different days of the same week were averaged. All behaviours combined and all harmful behaviours (tail or ear manipulation, manipulation of other body parts and belly-nosing) were analysed in addition to each individual behaviour. Biting, rooting and other contact with the enrichment devices were combined and analysed as a single value for interaction with the enrichment. The unit for analysis was pen (n = 48).

Average daily gain (ADG) and average daily feed intake (ADFI) were analysed separately in the weaner and finisher stages and overall for the same reason mentioned above for lesion scores and behaviours. Batch, enrichment in the weaner stage and the dietary treatment were included as the fixed effects for the weaner ADG and ADFI. The finisher ADG, ADFI and carcass weight further included enrichment treatments in both stages and all treatment interactions as fixed effects. The random effect of block was also included.

**Table 3. Total number of pigs removed as tail biting victims or biters, treated by injection[a] for tail biting injury, and the number of tail biting outbreaks recorded in different treatment groups.** If a pig was removed repeatedly, the number was counted repeatedly in order to reflect the repeated tail biting events.

| | Removed as victims[b] | Removed as biters | Injected[a] in home pens | Number of outbreaks |
|---|---|---|---|---|
| Diet[c] | | | | |
| High | 89 (75) | 17 | 29 | 14 |
| Standard | 60 (46) | 19 | 20 | 12 |
| Subtotal | 149 (121) | 36 | 49 | 26 |
| Weaner enrichment | | | | |
| Toy | 60 (52) | 15 | 27 | 8 |
| Wood | 86 (53) | 28 | 22 | 12 |
| Finisher enrichment | | | | |
| Toy | 26 (16) | 10 | 1 | 4 |
| Wood | 22 (12) | 5 | 2 | 2 |
| Overall enrichment | | | | |
| Toy—Toy | 32 (18) | 9 | 14 | 4 |
| Toy—Wood | 45 (29) | 11 | 15 | 6 |
| Wood—Toy | 65 (50) | 25 | 14 | 11 |
| Wood—Wood | 52 (36) | 13 | 9 | 5 |
| Subtotal | 194 (133) | 58 | 52 | 26 |

[a] Subcutaneous injections of antibiotics (Norocillin®, Norbrook) and analgesics (Loxicom®, Norbrook).

[b] As described in [25], pigs were removed for tail biting outbreak interventions and also for medical treatments in pens without reaching the tail biting outbreak criteria whenever necessary. Numbers in brackets are pigs removed as victims as part of the tail biting outbreak protocol.

[c] The dietary treatment began in week 4 post-weaning and therefore the numbers prior to this time were excluded.

Measurements taken from both the wooden posts and the floor toys were analysed as weight loss (kg) per day per pig. Batch, week and treatment were included as fixed effects; week was used as the repeated effect with block as the random effect. All results are presented hereafter as least squares means ± standard error of the mean. For behavioural parameters which were transformed, the least squares means ± standard error of the original values were reported and indicated.

## Results

During the course of the experiment, 26 tail biting outbreaks were recorded. Table 3 lists the number of pigs removed or treated in the pen for infected tails (by injection of antibiotics and analgesics) due to severe tail biting. Nine pigs were permanently removed from the experiment due to experiencing a severe tail lesion. In total, 161 individual pigs had tail wounds that necessitated temporary removal from their home pens for treatment (of which 20 pigs were removed twice, five pigs removed three times and one pig removed four times), and 58 individual pigs were removed as tail biters. An additional 52 pigs were treated with antibiotic injection in the home pen. At the end of the experiment, 66.9% of pigs had some level of tail amputation. The likelihood of having an outbreak did not differ between treatments.

Pens with pigs on the standard fibre diet had fewer pigs removed on average (3.07 ± 0.37) than pigs on the high fibre diet (4.34 ± 0.44; $F_{1,43}$ = 5.09, p = 0.03). Pens which had the rubber floor toy during the weaner stage tended to have fewer pigs removed for treatment at some point (3.16 ± 0.38) than those with the spruce post (4.22 ± 0.43) but this was not statistically significant ($F_{1,43}$ = 3.58, p = 0.07). Looking at the finisher stage alone, there was an effect of the interaction between enrichment and diet: pens receiving wood in the finisher stage with the standard fibre (standard-wood) had lower number of pigs removed or treated for tail injuries

with antibiotic injections ($0.44 \pm 0.19$, compared to standard-toy $1.67 \pm 0.42$ and high-wood $1.85 \pm 0.43$, $F_{1,41} = 10.42$, $p < 0.01$).

## Physical scores

The overall tail lesion scores (damage + blood) did not differ between pigs receiving different dietary or enrichment treatments. However, a tendency was found for a higher proportion of uninjured pigs (overall score 0) in the finisher stage when pigs received wood during the finisher stage (wood $0.235 \pm 0.015$ vs. toy $0.197 \pm 0.015$, $F_{1,69.7} = 3.7$, $p = 0.06$). When the tail damage score was analysed independently of the blood score, pigs which had a toy as enrichment during the weaner stage tended to have a lower proportion of pigs which scored 2–3 for tail damage in the finisher stage, compared to those who had wood ($0.015 \pm 0.006$ vs. $0.033 \pm 0.011$, $F_{1,87.84} = 3.56$, $p = 0.06$). Pens given the standard fibre diet also tended to have a higher proportion of pigs which scored 0 for tail damage during the finisher stage than those given the high fibre diet (standard $0.292 \pm 0.017$ vs. high $0.254 \pm 0.017$, $F_{1,67.7} = 3.21$, $p = 0.08$). No difference was found in ear lesion score.

## Behaviour

Overall, tail biting was more frequently observed in the high fibre than the standard fibre pigs ($p < 0.05$, Table 4). However, this was only significant overall and in the weaner stage ($p < 0.05$), but not in the finisher stage (Table 4). There was no effect of enrichment on all harmful behaviours across stages, but in the weaner stage alone, weaner pigs given wood ($p < 0.05$; Table 4) or the high fibre diet ($p < 0.05$, Table 4) showed more harmful behaviours.

**Table 4. Treatment effects on behaviour in different production stages.** Behaviours were analysed as frequency/pig/min (LS mean ± s.e.).

| | Enrichment | | | | | | Dietary fibre level | |
| --- | --- | --- | --- | --- | --- | --- | --- | --- |
| | Weaner | | Finisher | | Overall[a] | | | |
| | Toy | Wood | Toy | Wood | Toy | Wood | Standard | High |
| Tail biting[b] | | | | | | | | |
| Overall | 0.011±0.001 | 0.014±0.001 | - | - | 0.011±0.001 | 0.014±0.001 | 0.010±0.001* | 0.015±0.001* |
| Weaner | 0.019±0.002 | 0.024±0.002 | - | - | - | - | 0.018±0.002* | 0.025±0.002* |
| Finisher | 0.006±0.002 | 0.008±0.002 | 0.006±0.002 | 0.008±0.002 | - | - | 0.006±0.002 | 0.008±0.002 |
| All harmful[c] | | | | | | | | |
| Overall | 0.031±0.010 | 0.034±0.010 | - | - | 0.030±0.010 | 0.035±0.010 | 0.030±0.010 | 0.035±0.010 |
| Weaner | 0.042±0.013* | 0.051±0.013* | - | - | - | - | 0.042±0.013* | 0.051±0.013* |
| Finisher | 0.029±0.003 | 0.026±0.003 | 0.027±0.003 | 0.028±0.003 | - | - | 0.027±0.003 | 0.028±0.003 |
| Enrichment interaction[d] | | | | | | | | |
| Overall | 0.099±0.004*** | 0.075±0.004*** | - | - | 0.109±0.004*** | 0.067±0.004*** | 0.091±0.004 | 0.085±0.004 |
| Weaner | 0.118±0.006*** | 0.067±0.006*** | - | - | - | - | 0.093±0.006 | 0.092±0.006 |
| Finisher | 0.083±0.006 | 0.082±0.006 | 0.098±0.006*** | 0.067±0.006*** | - | - | 0.088±0.006 | 0.077±0.006 |

[a] When analysing the overall effect of the enrichment, only the enrichment at the time of recording was used as the fixed effect.

[b] Tail biting was transformed using square root and the values reported hereof are the LS mean ± s.e. of the original values.

[c] All harmful = tail biting + ear biting + biting other parts of the body + belly-nosing.

[d] Enrichment interaction = bite + root + other contact with the device.

Significant differences between two columns of the treatment within the same stage are indicated by

* $p < 0.05$ and

*** $p < 0.001$.

No difference was found in the interactions between treatments.

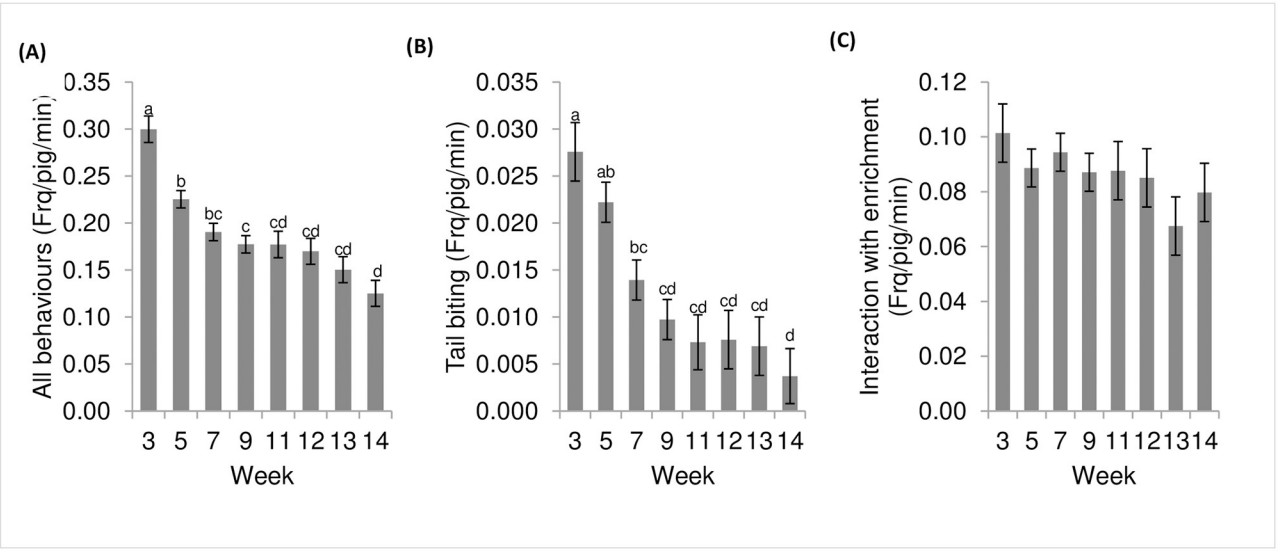

**Fig 2. Frequency of selected behaviours across time (frequency/pig/min (LSmean ± s.e.)).** (A) All recorded behaviours combined (Table 2, p < 0.001) (B) Tail biting (p < 0.001 with square root transformation, the values presented in the graph are the original values.) (C) All interactions with the enrichment (bite + root + other contact with the device, N.S.).

The overall level of interaction with the enrichment was affected by enrichment treatment. Pigs had more overall interactions with the rubber floor toy than wood (p < 0.001, Table 4). The same effect was found in the two stages separately (Table 4).

Pigs receiving the standard diet performed more aggressive behaviours near the feeder than pigs fed the high fibre treatment (0.019 ± 0.002 vs. 0.012 ± 0.002; $F_{1,52}$ = 7.26, p < 0.01, the parameter was square root transformed and the LS means ± s.e. of the original values presented). No effect was found regarding ear biting or other harmful behaviours individually, and neither was there difference in play behaviours between treatments.

There was an effect of time (week) on most behaviours, with a declining trend in frequency of observation as the pigs grew older, except for the interaction with the enrichment (Fig 2A–2C).

## Growth

Pigs fed the standard diet had a higher average daily gain (ADG) in the finisher stage than pigs fed with the high fibre diet (1.09 ± 0.01 kg vs. 1.06 ± 0.01 kg; $F_{1,641}$ = 5.55, p < 0.05). There was no effect of diet on the weaner ADG. Similar to the growth rate in the finisher stage, the standard diet contributed to a heavier carcass weight (74.24 ± 0.70 kg) than the high fibre diet (72.48 ± 0.71 kg; $F_{1,637}$ = 6.6, p < 0.05). The average overall daily feed intake was also higher in pigs fed with the standard diet (1.81 ± 0.01 kg/pig/day) than with the high fibre diet (1.75 ± 0.01 kg/pig/day, $F_{1,41.2}$ = 9.42, p < 0.01). No difference in growth was found between any of the enrichment treatments.

## Enrichment measurement

The rate of wear of the wooden post (reduction in length or weight) and the floor toy (reduction in weight) did not differ between enrichment or dietary treatments. Both enrichment items showed an increased rate of weight loss as pigs grew older.

## Discussion

This study compared the effectiveness of two point-source environmental enrichment items, combined with a standard or high fibre diet in preventing tail biting behaviour and tail lesions in undocked pigs kept on fully-slatted floors. The main finding was that none of the combined or individual treatments were effectively at prevented tail biting, and there were few differences between them across all measurements. A substantial number of pigs (161 out of 672) needed to be removed from their home pens either for tail treatments or for outbreak intervention measures, and the prevalence of partial tail amputation caused by biting injury reached 66.9%. The high fibre diet did not reduce tail biting behaviour, nor tail lesions, relative to the standard diet, and the benefits of the enrichment devices, although statistically significant at times, were marginal in terms of their effectiveness. Moreover, their effects were specific to different growth stages of the pigs.

Contrary to the original hypothesis, pigs fed a higher fibre diet did not perform less tail biting, especially during the weaner stage, and more pigs were removed from pens provided with the high fibre diet during this stage. Similar results were found with regard to the lesion scores; pigs in the high fibre treatment tended to have higher tail damage scores during the finisher stage. According to de Leeuw et al [21], a high fibre content in the diet reduced restlessness in sows. Thus, we hypothesised that the high fibre would reduce activity levels after feeding. However, sows are usually restricted-fed, whereas in this study the pigs were fed *ad libitum*. In addition, sows are more capable of fermentation of fibre in the hindgut than growing pigs, so for both of these reasons the effect of fibre may not be the same. A study showed that increased dietary fibre only had an effect on decreasing restrictedly-fed growing pigs' physical activity when the pigs were housed in a straw-bedded environment [30]. They concluded that under barren housing conditions, pigs' activity levels were more affected by the lack of rooting materials and therefore the inability to satiate their motivation to root, rather than the diet itself.

In fact, in the present study, the high level of fibre in the diet increased tail-directed behaviours and tail lesions. One reason could be that pigs' faeces were affected by the high fibre diet. Oliviero et al [31] found that sows fed with higher fibre diets had softer faeces and higher water consumption. Soft and liquid faeces can more easily attach to the hindquarters of the pigs, which may stimulate increased levels of exploration from other pen mates. Moreover, Kallabis and Kaufmann [22] reported that pigs fed with a more fibrous diet tended to have longer feeding bouts. In the context of the current experiment, where pigs were housed on slatted floors with limited environmental enrichment and a single-spaced feeder (which are known risk factors for tail biting [32]), longer feeding bouts might have increased the likelihood of tail biting incidence. Another possible explanation could be the difference in energy between the standard and high fibre diets was greater in the finisher stage (Table 1). A study suggested a link between a low energy diet and a higher risk of tail biting, as pigs on a low energy diet slightly preferred to chew on blood-soaked ropes than pigs fed with a control diet [33].

However, pigs fed with the high fibre diet had fewer aggressive encounters at the feeder. One of the advantages of a higher fibre content in the diet is a reduction in the motivation to feed, and it can contribute to fewer meals per day [22]. Although this might have reduced the competition for feed, another consequence could be that the high fibre might also prolong time spent at each meal, which would increase rather than reduce competition. The feed intake was indeed lower in pigs fed the high fibre diet in this study, but it should also be noted that although the primary difference between the standard and high fibre diet was the level of fibre content, the ingredients used in each diet differed slightly (e.g. a higher percentage of vegetable oil in high fibre diet, Table 1). The different composition of the diets could influence their taste and palatability, and therefore alter pigs' feeding behaviours. More detailed recordings of pigs'

feeding behaviour are needed to further investigate the link between high dietary fibre and damaging behaviours.

Higher fibre also resulted in a lower average daily gain, which is similar to what a previous study found [22]. The difference in weight gain was only significant in the finishers possibly due to the fact that weaners in the current study had standardised starter and link diet for a total of 3 weeks post-weaning and were only put on treatment diets for 4 weeks before being transferred to the finisher accommodation. Similar results were reported when a high fibre diet was provided to weaners from 4 to 7 weeks of age [23]. The relatively shorter period of dietary treatment in the weaner stage and the time needed to adjust to a novel diet for both standard and high fibre treatment may explain why no clear effect on growth was found.

In terms of enrichment provision, weaners receiving wood tended to have higher tail lesion scores, and more weaners were removed due to biting from the wood enriched pens. On the contrary, pigs which had wood enrichment as finishers fed with the standard diet had fewer pigs affected by tail biting per pen (removed for treatment, removed as biters or injected with antibiotics). These results suggested that when only a single enrichment device was available, the rubber floor toy for weaners and the spruce post for finishers were more effective in reducing lesion scores. Docking et al [34] also showed that pigs at different ages used enrichment items differently. An extensive review of enrichment for pigs reported that pigs' interest in specific objects could change over time based on their own behavioural development [35]. The rubber floor toy had soft, chewable arms, and it was loose on the floor area, which may have made it more readily accessible for weaners to manipulate than the spruce wooden post, which was harder and might require more effort, skill, and a stronger mouth to chew and use. Enrichment provision should be based on the properties suitable for different ages of the pigs, and appropriate enrichment provision would have a positive effect to reduce injuries from harmful behaviour.

Nevertheless, the overall impact of tail biting in this study was severe, and the effectiveness of either of these single-point enrichment items was marginal. To our knowledge, no study to date has yet demonstrated the possibility of managing tail biting in undocked pigs without loose rooting materials provided on the floor or via dispensers [36]. Frustration from the inability to satisfy the natural motivation to explore and forage due to inadequate materials is a major cause of tail biting in pigs [37,38]. Bracke [39] proposed that for an environmental enrichment to be considered "proper," it should provide occupation and be able to prevent abnormal behaviours. The basic enrichment provision used in this study failed that test. However, enrichment of this type is still common on many commercial farms, which was the reason why this level of provision was used. Indeed, in Irish commercial units where point source enrichment items are the most commonly used [18], there is normally a much higher pigs to enrichment ratio, than in this study [40]. The current study found that pigs interacted with the enrichment items, and that this interaction did not decrease over time, suggesting they have some occupation value beyond novelty to the pigs. Nevertheless, the enrichment items were still insufficient to stop intact-tailed pigs from tail biting. This might also explain why although in the finisher stage a higher interaction with the floor toy was recorded, the tail lesions were not significantly improved, and the increased use of the enrichment did not contribute to less frequent tail biting either.

Despite the numerous tail biting outbreaks recorded, overall tail lesion scores in the current experiment were lower than expected. In fact they were lower than those recorded in another study with undocked pigs, which recorded no tail biting outbreak [36]. The low tail lesion scores are likely due to the timing of the scoring: pigs were scored every 2 weeks, and based on the experimenter's experience, tail lesions tend to heal quickly. Based on a previous study on the histopathology of tail docking, superficial healing of the tail injury could take 3–7 days

depending on the severity [41]. Therefore, the scores likely did not reflect lesions at their worst, immediately post-biting, which is why they underestimated the severity of the biting. Moreover, the tail scoring system used in the current experiment is likely more sensitive at distinguishing the severity of fresh lesions, rather than older lesions. Furthermore, tail biting events peaked between 3–7 weeks post-weaning in the current study (during which time only two out of a total of seven scorings were conducted), hence the average tail score over time could have also resulted in an underestimation and was less indicative of the actual severity of tail biting, compared to tail amputation score or the number of pigs removed and treated.

In conclusion, this study showed that simply increasing dietary fibre level and providing a single point-source enrichment at a 14:1 pig to enrichment ratio was not enough to control tail biting in pigs with intact tails, even if the enrichment item was used frequently by pigs. Severe tail biting was prevalent across all treatments and proved difficult to control. Pigs provided with the higher fibre diet tended to have higher tail lesion scores and performed more tail biting behaviour. The enrichment item which we considered suitable for use in the different stages (floor toy for weaners and spruce post for finishers) was slightly more effective than its counterpart in alleviating the severity of tail lesions, although neither item succeeded in reducing the risk of tail biting. More advanced strategies taking into consideration the complex factors behind tail biting are needed to manage tail biting in undocked pigs on fully-slatted systems.

## Supporting information

**S1 Data.**
(XLSX)

## Acknowledgments

We would like to acknowledge the help and support from the staff at the Pig Research Facility in Moorepark, Teagasc, Ireland, intern students Alexandra Courty and Marleen van de Heide, and also Dr Edgar Garcia Manzanilla for the advice on feed formulation.

## Author Contributions

**Conceptualization:** Jen-Yun Chou, Keelin O'Driscoll, Dale A. Sandercock, Rick B. D'Eath.

**Data curation:** Jen-Yun Chou.

**Formal analysis:** Jen-Yun Chou.

**Funding acquisition:** Keelin O'Driscoll, Dale A. Sandercock, Rick B. D'Eath.

**Investigation:** Jen-Yun Chou.

**Methodology:** Jen-Yun Chou, Keelin O'Driscoll, Dale A. Sandercock, Rick B. D'Eath.

**Project administration:** Keelin O'Driscoll, Dale A. Sandercock, Rick B. D'Eath.

**Supervision:** Keelin O'Driscoll, Dale A. Sandercock, Rick B. D'Eath.

**Validation:** Jen-Yun Chou, Keelin O'Driscoll, Dale A. Sandercock, Rick B. D'Eath.

**Visualization:** Jen-Yun Chou.

**Writing – original draft:** Jen-Yun Chou.

**Writing – review & editing:** Keelin O'Driscoll, Dale A. Sandercock, Rick B. D'Eath.

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
