## [Decision Letter · Decision Letter 0]

3 Sep 2020

PONE-D-20-19470

Can increased dietary fibre level and a single enrichment device reduce the risk of tail biting in undocked growing-finishing pigs on fully slatted systems?

PLOS ONE

Dear Dr. Chou,

Thank you for submitting your manuscript to PLOS ONE. After careful consideration, we feel that it has merit but does not fully meet PLOS ONE’s publication criteria as it currently stands. Therefore, we invite you to submit a revised version of the manuscript that addresses the points raised during the review process.

In addition to the detailed comments provided by the two reviewers, which you find at the end of this message, please consider the following issues arising from my own thorough reading of the manuscript:

Lines 44-45 If this Directive is still in force, please change to “lays down” and “is banned”

Line 52 Do you really want to say that tail docking** potentially** masks the underlying issues – I mean do you need the word “potentially” here?

Line 73 “the way an enrichment material is presented” rather than “the presentation of an enrichment material” (which is ambiguous, can mean simply that it is presented)

Line 99 In my understanding, the term behaviour repertoire stands for the entire diversity of behaviours that an animal performs. I know not everyone thinks “a behaviour” is a grammatically correct term but it is widely used that way in the field.

Section Animals and housing Please include information on how and what the sows were fed

References Please carefully proof read your references and make sure they are consistently formatted and in agreement with the guidelines for PLOS ONE.

We look forward to receiving your revised manuscript.

Kind regards,

I Anna S Olsson, Ph.D.

Academic Editor

PLOS ONE

Journal Requirements:

Reviewers' comments:

Reviewer's Responses to Questions

**Comments to the Author**

1. Is the manuscript technically sound, and do the data support the conclusions?

Reviewer #1: Yes

Reviewer #2: Partly

2. Has the statistical analysis been performed appropriately and rigorously? 

Reviewer #1: Yes

Reviewer #2: Yes

3. Have the authors made all data underlying the findings in their manuscript fully available?

Reviewer #1: Yes

Reviewer #2: Yes

4. Is the manuscript presented in an intelligible fashion and written in standard English?

Reviewer #1: Yes

Reviewer #2: Yes

5. Review Comments to the Author

Reviewer #1: The manuscript is well written and with a high level of detail, thus easily understandable. The manuscript concerns the investigation of the effectiveness of dietary fiber and a single point-source enrichment to reduce the occurrence of tail biting behaviour and tail lesions in undocked pigs on fully slatted floors. The subject is highly relevant for the research field, despite that the study found neither dietary fiber nor point-source enrichment to be effective. My only concern is with the statistical analysis where more detail is needed to be able to replicate the analyses. This and a few other concerns are specified in the comments below.

L211-217: I understand the benefits of direct observations especially when considering the high detail you put into enrichment engagement, which would probably have been hard to watch on video. But didn’t you ever come into trouble with multiple behaviour types being shown at the same time, making it impossible for the observer to record everything and not miss anything? Was this possible due to the long bout criteria? Perhaps justify this a bit more in the text. Also, as you observed 5 minutes per pen, did you in any way allow for the pigs acclimatize to the observer?

Table 2: The descriptions for tail manipulation and aggression should include that these behaviour do not occur at the feeder or while feeding, and for aggression also that this does not occur around the enrichment; to make them mutually exclusive from the two other behaviour described.

231-232: What was the reason that you chose the rather “simple” Chi-squared test? When you use a mixed model for other parameters.

L233: did you use a Poisson distribution as you work with counts? Or perhaps negative binomial? Please specify.

L235: you here mention replicate for the first time – is that the same as the batch you mentioned earlier? Please use the same word.

L237: was this then with a Gaussian/normal distribution? Please specify. I am lucky that I know SAS (I use R), but otherwise I would not understand the Glimmix and Proc Mixed terms. Thus, these terms are not so important, but specifying the used distribution is important for me to replicate the analysis.

L237-243: please specify the analysis for each parameters. Otherwise it will not be possible to replicate your analysis if needed.

L242-243: poisson distribution is usually used for count data. Were the data originally count data? Or how do you justify using this distribution?

255-256: what the scores 2 and 3 combined for both tail lesions and ear lesions?

Table 3 Caption: should tail bittin victims be replaced by tail biting victims?

L429: delete the second word ‘as’.

Reviewer #2: This paper describes a study in which the relative potential of different kinds of enrichment and high dietary fibre to reduce tail biting (damage) was evaluated in undocked pigs. The majority of research on tail biting has been carried out with docked pigs and as a result, such research does not necessarily give proper insight into the efficacy of specific treatments for undocked pigs. This insight is very much needed as the EU is increasingly moving towards the production of undocked pigs, and in all likelihood this will happen whilst slatted floor systems are still the norm. Therefor this paper makes an important contribution to the research on this significant welfare problem.

The study and its analysis seem carefully executed and, for the most part, are clearly described in this paper. In several places the authors state that enrichment & additional fibre weren’t was effective enough to control tail biting. It would be good if it was made explicit what the authors actually mean by ‘control’ (e.g., completely prevent? Limit to x% of the pigs getting damage)? This is especially important as the statement could be misinterpreted to mean that the enrichment treatments did not have a beneficial effect at all, which can’t be gauged from the study due to an absence of a control without enrichment. In fact, in some places the authors actually directly make this claim, which is incorrect and should be amended. Also, I think it would be good to draw the statement on the lack of control into a broader context, as not only the enrichment, but also a close monitoring system combined with a dedicated removal protocol was in place, as well as additional emergency enrichment when problems occurred. Even all this combined did not lead to an elimination of tail biting behaviour.

Some more detailed comments are described below.

Title: I’d say that the pigs are in fully slatted systems rather than on? Please review

L25: Whereas the phrasing “pigs with intact tails in a conventional fully-slatted flooring system” suggest to me that the pigs were actually inside the floor. Again, please review and adapt if necessary.

L25-29: This sentence is very long and it’s not clear here how the enrichment treatments are actually applied. Please rephrase.

L49: “fear of pig producers” I understand what you mean here of course but still the phrasing is confusing (as if these are pigs fearing their producer). Please consider revising.

L62: Change “difficult” to “especially difficult” as difficulties are not limited to slatted systems only

L67-78: It needs to be stated more clearly in the introduction why wood was chosen as enrichment, knowing from previous research on this topic that it has little effect on tail biting. Also, rubber toys (and their previously reported effects, or the lack thereof) should be introduced similar to what was done for wood.

L104: please remove the comma

L114: Were the 5x daily checks also performed during weekends?

L117: Please state more clearly if the pigs were only removed for treatment (i.e., injecting / spraying them) or for a longer stay in a hospital pen. Also, define more clearly what you mean by ‘ethical reasons’

L120: Outbreak occurrence seems to be a primary measure here, so it should be clearly defined in the current paper, not just by referring to a previous one

L146: Please add “per pen”

L166: ‘middle of the pen’ why is this relevant? I don’t assume it stayed there as the pigs could move it about?

L168: Please provide the measures for the height and width separately, rather than the perimeter as these would affect whether a pig can put it mouth around the wood, manipulating it more effectively. Or was the wood cylindrical? In that case please state so.

206-209: Is there any evidence to support that a tail that is amputated closer to the body represents either a greater welfare problem or more prolonged chewing? If not, I don’t see the validity of this scale. If there is evidence please present it in the paper. If there is no evidence but you were working under the assumption that this is true, please state so instead.

L214: Please describe more clearly what you mean by ‘bout criterion’

L219: ‘tail-in-mouth behaviour’ is not a proper descriptor as it needs to be define itself. Please adapt the description to say what you actually observed.

For the definition of fighting, describe when you would consider a behaviour to be ‘play fighting’.

L227: Please change ‘enrichment’ to ‘the enrichments’

L250-251: I’m surprised to see week as the repeated measure here. I would assume that you used repeated observations on the same pen (conducted in different weeks) as repeated measures. To me it seems that what you are describing is that instead you considered observations conducted in the same week (but on different pens) as repeated. Please clarify.

L272: I assume that, where relevant (e.g. poisson based models) these were backtransformed? Please state so clearly.

L276: Remove ‘from each treatment group’ to increase clarity.

L308: proportion or %? (there’s a % sign later on)

L331: The table also features ***, please define

L347-354: Please describe the presence or absence of effects of enrichment on growth.

L356-358: Please provide full analysis details for this measure.

L360: “This study investigated the effectiveness of a single point-source environmental enrichment”. This statement is false, you compared the effectiveness of two types of enrichment. To do what you said you would need to contrast your enrichment groups to control groups without enrichment. Please adapt the phrasing. Same issue in lines 367 and 434. Although I agree with you that a lot of animals sustained damage in this study (showing that the enrichment doesn’t entirely solve the problem to a sufficient extent), there is no way of telling whether your wood and rubber toy weren’t improving things but just to an equal extent.

6. PLOS authors have the option to publish the peer review history of their article (what does this mean?). If published, this will include your full peer review and any attached files.

Reviewer #1: No

Reviewer #2: No

---

## [Author Response · Author response to Decision Letter 0]

16 Oct 2020

Editor’s comments

Dear editor, we would like to thank you for your time to have thoroughly read and considered our manuscript. We have responded to all your and both reviewers’ comments below. In this rebuttal letter, all our responses are in blue/italic after each comment. We indicated the line numbers where changes were made, and the line numbers used were based on the track changes manuscript (“all markup” mode) to show specifically how we revised the manuscript. We are happy to see the positive comments from both reviewers, agreeing that this manuscript has its merit in contributing the current literature in the understanding of tail biting and its possible solutions. We hope you could further consider our revised manuscript and we will sincerely await any further comments. 

Lines 44-45 If this Directive is still in force, please change to “lays down” and “is banned”

Response: L49-50 amended as suggested.

Line 52 Do you really want to say that tail docking potentially masks the underlying issues – I mean do you need the word “potentially” here?

Response: “Potentially” removed (L56).

Line 73 “the way an enrichment material is presented” rather than “the presentation of an enrichment material” (which is ambiguous, can mean simply that it is presented)

Response: L74 amended as suggested.

Line 99 In my understanding, the term behaviour repertoire stands for the entire diversity of behaviours that an animal performs. I know not everyone thinks “a behaviour” is a grammatically correct term but it is widely used that way in the field.

Response: “Repertoire” deleted (L105).

Section Animals and housing Please include information on how and what the sows were fed

Response: More information added in L153-156.

References Please carefully proof read your references and make sure they are consistently formatted and in agreement with the guidelines for PLOS ONE.

Response: We apologise for the oversight and errors in reference formatting and have corrected and proof-read all references to make sure they adhere to the PLOS ONE guidelines.

Reviewer #1 

The manuscript is well written and with a high level of detail, thus easily understandable. The manuscript concerns the investigation of the effectiveness of dietary fiber and a single point-source enrichment to reduce the occurrence of tail biting behaviour and tail lesions in undocked pigs on fully slatted floors. The subject is highly relevant for the research field, despite that the study found neither dietary fiber nor point-source enrichment to be effective. My only concern is with the statistical analysis where more detail is needed to be able to replicate the analyses. This and a few other concerns are specified in the comments below.

We would like to thank the reviewer’s acknowledgement and we addressed the comments below line by line and revised the manuscript accordingly. The line numbers referred below are based on the track changes manuscript (“all markup” mode) to indicate clearly where changes were made.

L211-217: I understand the benefits of direct observations especially when considering the high detail you put into enrichment engagement, which would probably have been hard to watch on video. But didn’t you ever come into trouble with multiple behaviour types being shown at the same time, making it impossible for the observer to record everything and not miss anything? Was this possible due to the long bout criteria? Perhaps justify this a bit more in the text. Also, as you observed 5 minutes per pen, did you in any way allow for the pigs acclimatize to the observer?

Response: The observer was thoroughly trained in conducting direct behaviour observation in more challenging conditions (on commercial farms with more pigs in a pen) in previous studies. In addition, practice sessions were conducted before the actual observation. Of course, it is not realistic to say everything was recorded without missing anything, but as the observation was conducted by one observer and all pens were observed and recorded in a consistent way, we feel confident saying that the methodology is trustworthy. And yes, the observer did acclimatise the pigs by walking up and down the passageway and always vocally “communicate” with the pigs so that they know her presence. Due to the long time the experimenter (observer) spent with the pigs (inside and outside the pen), they became very used to the observer after a few weeks into the study and never showed a startle reaction when the observer was present. We have included some additional information about this in the text (L243-248).

Table 2: The descriptions for tail manipulation and aggression should include that these behaviour do not occur at the feeder or while feeding, and for aggression also that this does not occur around the enrichment; to make them mutually exclusive from the two other behaviour described.

Response: Table 2 amended as suggested.

231-232: What was the reason that you chose the rather “simple” Chi-squared test? When you use a mixed model for other parameters.

Response: We used Chi-squared test because the number of outbreaks was one simple variable over the whole study and by using Chi-squared test we can readily compare between treatment groups. Other measures we took were all repeated measures including random effects and therefore we used more complex models.

L233: did you use a Poisson distribution as you work with counts? Or perhaps negative binomial? Please specify.

Response: We used Poisson distribution and L266 amended.

L235: you here mention replicate for the first time – is that the same as the batch you mentioned earlier? Please use the same word.

Response: Yes, replicate is the same as batch. We corrected the terms to make it consistent (L268, 280, 302, 307).

L237: was this then with a Gaussian/normal distribution? Please specify. I am lucky that I know SAS (I use R), but otherwise I would not understand the Glimmix and Proc Mixed terms. Thus, these terms are not so important, but specifying the used distribution is important for me to replicate the analysis.

Response: L270 amended.

L237-243: please specify the analysis for each parameters. Otherwise it will not be possible to replicate your analysis if needed.

Response: We have indicated which measures were transformed using square root to make it clearer (L271, L274-279). Other analysis information was detailed in the following paragraphs in this section.

L242-243: poisson distribution is usually used for count data. Were the data originally count data? Or how do you justify using this distribution?

Response: The data in question were the proportions of tail damage score 2 & 3 (L276-278), so it was a form of count data as well. After rechecking, we thought the most appropriate distribution was binomial distribution with a logit link function. The results given were the same with slightly different numbers and we edited all relevant parts. (L276-279, L353-354)

255-256: what the scores 2 and 3 combined for both tail lesions and ear lesions?

Response: Yes, both scores were combined. (L292-293 clarified) 

Table 3 Caption: should tail bittin victims be replaced by tail biting victims?

Response: Table 3 caption amended as suggested (L324).

L429: delete the second word ‘as’.

Response: “As” deleted (L476).

Reviewer #2

This paper describes a study in which the relative potential of different kinds of enrichment and high dietary fibre to reduce tail biting (damage) was evaluated in undocked pigs. The majority of research on tail biting has been carried out with docked pigs and as a result, such research does not necessarily give proper insight into the efficacy of specific treatments for undocked pigs. This insight is very much needed as the EU is increasingly moving towards the production of undocked pigs, and in all likelihood this will happen whilst slatted floor systems are still the norm. Therefor this paper makes an important contribution to the research on this significant welfare problem.

The study and its analysis seem carefully executed and, for the most part, are clearly described in this paper. In several places the authors state that enrichment & additional fibre weren’t was effective enough to control tail biting. It would be good if it was made explicit what the authors actually mean by ‘control’ (e.g., completely prevent? Limit to x% of the pigs getting damage)? This is especially important as the statement could be misinterpreted to mean that the enrichment treatments did not have a beneficial effect at all, which can’t be gauged from the study due to an absence of a control without enrichment. In fact, in some places the authors actually directly make this claim, which is incorrect and should be amended. Also, I think it would be good to draw the statement on the lack of control into a broader context, as not only the enrichment, but also a close monitoring system combined with a dedicated removal protocol was in place, as well as additional emergency enrichment when problems occurred. Even all this combined did not lead to an elimination of tail biting behaviour.

We would like to thank the reviewer’s recognition of the importance of our manuscript and we replied to the comments below line by line and revised the manuscript accordingly. The line numbers referred below are based on the track changes manuscript (“all markup” mode) to indicate clearly where changes were made.

We agreed with the reviewer that more careful conclusion should be drawn in terms of our enrichment treatments, in order not to mislead some readers that enrichment has no benefit at all, and we have clarified several such statements in the abstract (L40-46) and discussion (L406-410) to be more specific. Regarding the lack of “negative control” in a broader context, we also included some clarifications in the methods section (L193-197). However, we would also like to raise the point that inappropriate environmental enrichment could bring negative consequences, especially to tail biting, as shown in our previous studies and mentioned by some other authors as well. The minimal enrichment we provided in the current study, although sadly still prevalent in many countries worldwide, was shown to be insufficient to satisfy pigs’ foraging and inquisitive behaviours. As there is no clear definition of what percentage of tail biting is “acceptable” or not, it is indeed difficult to draw a definitive conclusion on if certain treatments, e.g. the ones we have provided, actually “control” or “improve” tail biting in pigs. We rechecked our claims and made sure overstatements were removed and when we talked about “effectiveness” and “control” of tail biting, we have qualifier sentences followed, to refer back to our actual outcomes (L33, L40-41, L43-46, L406-410, L410-413, L414-416, L470-472, L481-482, L513-515, L518-521).

Some more detailed comments are described below.

Title: I’d say that the pigs are in fully slatted systems rather than on? Please review

Response: L4 amended.

L25: Whereas the phrasing “pigs with intact tails in a conventional fully-slatted flooring system” suggest to me that the pigs were actually inside the floor. Again, please review and adapt if necessary.

Response: “Housing” added in L25.

L25-29: This sentence is very long and it’s not clear here how the enrichment treatments are actually applied. Please rephrase.

Response: L26-30 rewritten.

L49: “fear of pig producers” I understand what you mean here of course but still the phrasing is confusing (as if these are pigs fearing their producer). Please consider revising.

Response: L52 rephrased.

L62: Change “difficult” to “especially difficult” as difficulties are not limited to slatted systems only

Response: “Especially” added (L62).

L67-78: It needs to be stated more clearly in the introduction why wood was chosen as enrichment, knowing from previous research on this topic that it has little effect on tail biting. Also, rubber toys (and their previously reported effects, or the lack thereof) should be introduced similar to what was done for wood.

Response: More information added in L77-78 and L80-84.

L104: please remove the comma

Response: Comma removed (L109).

L114: Were the 5x daily checks also performed during weekends?

Response: On Sunday there were fewer checks due to the experimenter and staff taking time off (L119-120 clarified).

L117: Please state more clearly if the pigs were only removed for treatment (i.e., injecting / spraying them) or for a longer stay in a hospital pen. Also, define more clearly what you mean by ‘ethical reasons’

Response: More information added and clarified in L123-129. 

L120: Outbreak occurrence seems to be a primary measure here, so it should be clearly defined in the current paper, not just by referring to a previous one

Response: We have published the study focusing on the intervention protocol we used in the previous paper and due to the complexity of it, we feel it is better not to elaborate too much on it in case of distracting the readers. However, we included some additional information regarding the definition of outbreak in L132-136.

L146: Please add “per pen”

Response: “Per pen” added in L165.

L166: ‘middle of the pen’ why is this relevant? I don’t assume it stayed there as the pigs could move it about?

Response: L185-186 amended.

L168: Please provide the measures for the height and width separately, rather than the perimeter as these would affect whether a pig can put it mouth around the wood, manipulating it more effectively. Or was the wood cylindrical? In that case please state so.

Response: L187-188 clarified. In Figure 1 there is also a photo showing the actual wood post for the readers’ reference.

206-209: Is there any evidence to support that a tail that is amputated closer to the body represents either a greater welfare problem or more prolonged chewing? If not, I don’t see the validity of this scale. If there is evidence please present it in the paper. If there is no evidence but you were working under the assumption that this is true, please state so instead.

Response: Reference added in L229-230.

L214: Please describe more clearly what you mean by ‘bout criterion’

Response: L239-240 clarified.

L219: ‘tail-in-mouth behaviour’ is not a proper descriptor as it needs to be define itself. Please adapt the description to say what you actually observed.

For the definition of fighting, describe when you would consider a behaviour to be ‘play fighting’.

Response: More detailed description is added in the footnote “a” L251-252 of Table 2.

L227: Please change ‘enrichment’ to ‘the enrichments’

Response: L260 amended to “enrichment items”.

L250-251: I’m surprised to see week as the repeated measure here. I would assume that you used repeated observations on the same pen (conducted in different weeks) as repeated measures. To me it seems that what you are describing is that instead you considered observations conducted in the same week (but on different pens) as repeated. Please clarify.

Response: Our experimental unit was pen and on each pen observation was conducted four times every other week. As we average different observations in the same week on each pen, the repeated effect represents the observations conducted on each pen every other week repeatedly, and therefore week was the repeated effect.

L272: I assume that, where relevant (e.g. poisson based models) these were backtransformed? Please state so clearly.

Response: L309-311 clarified. (Also footnote “b” under Table 4 L369-370 and L381-382)

L276: Remove ‘from each treatment group’ to increase clarity.

Response: L315 amended.

L308: proportion or %? (there’s a % sign later on)

Response: It is proportion and the % was a typo and removed (L350).

L331: The table also features ***, please define

Response: The ** in the annotation should be *** which was a typo and corrected (L374).

L347-354: Please describe the presence or absence of effects of enrichment on growth.

Response: We included the effects of enrichment on growth in L399-400.

L356-358: Please provide full analysis details for this measure.

Response: The methods of recording this measure can be found in L260-262 and the analysis described in L270-272 “enrichment measurements” and L306-309. To make it clearer to the reader, some details were added in L402 and L403.

L360: “This study investigated the effectiveness of a single point-source environmental enrichment”. This statement is false, you compared the effectiveness of two types of enrichment. To do what you said you would need to contrast your enrichment groups to control groups without enrichment. Please adapt the phrasing. Same issue in lines 367 and 434. Although I agree with you that a lot of animals sustained damage in this study (showing that the enrichment doesn’t entirely solve the problem to a sufficient extent), there is no way of telling whether your wood and rubber toy weren’t improving things but just to an equal extent.

Response: L406-407 amended. As we previously replied to the reviewer about stressing the drawbacks of using “inappropriate” enrichment, we would argue to keep L414-416 and L481-482 as we did not say the enrichment was ineffective, but that its effectiveness was “marginal,” which was the outcome we observed in the current study with a high level of tail biting.

---

## [Editor Report · Decision Letter 1]

19 Oct 2020

Can increased dietary fibre level and a single enrichment device reduce the risk of tail biting in undocked growing-finishing pigs in fully slatted systems?

PONE-D-20-19470R1

Dear Dr. Chou,

We’re pleased to inform you that your manuscript has been judged scientifically suitable for publication and will be formally accepted for publication once it meets all outstanding technical requirements.

Kind regards,

I Anna S Olsson, Ph.D.

Academic Editor

PLOS ONE
---

## [Editor Report · Acceptance letter]

21 Oct 2020

PONE-D-20-19470R1 

Can increased dietary fibre level and a single enrichment device reduce the risk of tail biting in undocked growing-finishing pigs in fully slatted systems? 

Dear Dr. Chou:

I'm pleased to inform you that your manuscript has been deemed suitable for publication in PLOS ONE. Congratulations! Your manuscript is now with our production department. 

Kind regards, 

on behalf of

Dr. I Anna S Olsson 

Academic Editor

PLOS ONE